# The Distinctive Serum Metabolomes of Gastric, Esophageal and Colorectal Cancers

**DOI:** 10.3390/cancers13040720

**Published:** 2021-02-10

**Authors:** Zhenxing Ren, Cynthia Rajani, Wei Jia

**Affiliations:** 1Center for Translational Medicine, Shanghai Key Laboratory of Diabetes Mellitus, Shanghai Jiao Tong University Affiliated Sixth People’s Hospital, Shanghai 200233, China; rzhenxing2016@gmail.com; 2Cancer Biology Program, University of Hawaii Cancer Center, Honolulu, HI 96813, USA; 3School of Chinese Medicine, Hong Kong Baptist University, Kowloon Tong, Hong Kong, China

**Keywords:** metabolomics, gastric cancer, esophageal cancer, colorectal cancer, Warburg effect, metabolome

## Abstract

**Simple Summary:**

Cancer of the stomach, esophagus and colon are often fatal. Ways are being sought to establish patient-friendly screening tests that would allow these cancers to be detected earlier. Examination of the metabolomics results of cancer patient’s serum for certain metabolites unique for a particular cancer was the goal of this review. From studies conducted within the past five years several metabolites were found to be changed in cancer compared to non-cancer patients for each of the three cancers. Further confirmation of what was discovered in this review coupled with establishment of standard protocols may allow for cancer screening on patient blood samples to become routine clinical tests.

**Abstract:**

Three of the most lethal cancers in the world are the gastrointestinal cancers—gastric (GC), esophageal (EC) and colorectal cancer (CRC)—which are ranked as third, sixth and fourth in cancer deaths globally. Early detection of these cancers is difficult, and a quest is currently on to find non-invasive screening tests to detect these cancers. The reprogramming of energy metabolism is a hallmark of cancer, notably, an increased dependence on aerobic glycolysis which is often referred to as the Warburg effect. This metabolic change results in a unique metabolic profile that distinguishes cancer cells from normal cells. Serum metabolomics analyses allow one to measure the end products of both host and microbiota metabolism present at the time of sample collection. It is a non-invasive procedure requiring only blood collection which encourages greater patient compliance to have more frequent screenings for cancer. In the following review we will examine some of the most current serum metabolomics studies in order to compare their results and test a hypothesis that different tumors, notably, from EC, GC and CRC, have distinguishing serum metabolite profiles.

## 1. Introduction to Gastric Cancer

Gastric cancer (GC) is ranked third in cancer deaths world-wide. It is separated anatomically into either gastric adenocarcinomas (non-cardia GC) or gastro-esophageal-junction adenocarcinomas (cardia GC) and is further classified histologically into either diffuse or intestinal types [1,2]. Genomic profiling of primary non-cardia GC has led to the identification of four tumor subgroups: 9% positive for Epstein-Barr virus, 22% microsatellite unstable, 20% genomically stable and 50% chromosomally unstable tumors [1]. Each of these four types of tumors have specific histology and gene mutations associated with them (Figure 1A) [1]. *Helicobacter pylori* infection is a major cause of sporadic GC due to the contribution of the chronic inflammation induced by its colonization of the stomach [3]. True hereditary GC accounts for about 1–3% of all GC mainly due to mutations in Cadherin-1 (CDH1) and catenin alpha-1 (CTNNA1) mutations [4]. Environmental factors that are thought to promote GC include, low consumption of fruits and vegetables, high intake of salt, nitrates and pickled foods, smoking, obesity and gastro-esophageal-reflux (GERD) disease [5,6,7].

GC progression is often described by a sequence of events known as Correa’s Cascade (Figure 1B) [8]. The cascade begins with non-active gastritis (*H. pylori* negative) (NAG-) which then proceeds to chronic active gastritis (*H. pylori* positive) (CAG+) → precursor lesions of GC (PLGC) (atrophy, intestinal metaplasia, dysplasia) → GC [8,9]. The prognosis of GC is related to the stage at which it is diagnosed with the treatment being surgical removal of the cancer tissue and resection. Currently, diagnosis relies heavily on invasive techniques such as endoscopy with biopsy followed by pathological examination [10]. Early stages of GC are often asymptomatic [11] and therefore, the use of metabolomics to detect serum biomarkers that would allow an earlier diagnosis of GC has become an important area of ongoing research.

## 2. Introduction to Esophageal Cancer

Esophageal cancer (EC) is classified into either esophageal squamous cell carcinoma (ESCC) which arises from squamous epithelial cells in the esophagus or esophageal adenocarcinoma (EAC) which originates from glandular cells located near the stomach [16]. Esophageal squamous cell carcinoma (ESCC) is the most common type of esophageal cancer in Asian countries and it is ranked as the sixth leading cause of cancer death globally [17]. EAC is more prominent in the western world and is linked to obesity [18,19]. The most established risk factors for EC are smoking, alcohol and reflux esophagitis, however, they are not the only contributors [20]. In the United States, diets including fiber, vitamin B_6_, β-carotene and vitamin C were inversely correlated with esophageal cancer, while consumption of animal protein, cholesterol and vitamin B_12_ were directly associated with esophageal cancer [20,21].

EAC usually starts with the formation of Barret’s esophagus, a metastatic condition that involves replacement of the normal stratified squamous epithelium with simple columnar epithelium. There are several theories as to how this occurs which is explained in more detail in Figure 2A. The most current theory involves abnormal differentiation of gastric cardia progenitor cells that migrate into the distal esophagus [19,22]. The reversion to simple columnar epithelium makes the esophagus more resistant to chronic inflammation due to acid-peptic damage [19]. Acid, bile and caudal homeobox (CDX) genes contribute to the metaplasia. CDX genes can be expressed in response to conjugated bile acids (BAs), tumor necrosis factor-α (TNFα), deoxycholic acid(DCA), interleukin-1β (IL-1β), chenodeoxycholic acid (CDCA) and chronic acid exposure [23,24,25,26] GERD separates the junctions among squamous cells allowing exposure of progenitor cells to BAs, acid and inflammatory compounds which, in turn, induces the expression of CDX genes. Barrett’s esophagus eventually evolves into EAC (Figure 2A) [22,24,26].

ESCC is characterized by basal cell hyperplasia and dysplasia. The key initiator of ESCC is the transcription factor, SRY-box transcription factor 2 (Sox2) which has been found to cooperate with phosphorylated signal transducer and activator of transcription 3 (Stat3) to transform foregut basal keratin 5 positive progenitor cells (Figure 2B) [27,28]. Increased levels of phosphorylated Stat3 and Sox2 have been shown to closely correlate with poor outcomes in ESCC patients [27,28].

## 3. Introduction to Colorectal Cancer

Colorectal cancer (CRC) is the third most common cancer and the fourth most common cause of cancer-related death globally. It is strongly associated with western diet and lifestyle with higher incidence in North America and Europe [29,30]. CRC usually develops slowly (over 10 years) in an environment of chronic inflammation which is often initiated by environmental factors such as bacterial or viral infection. The infection activates an immune response that leads to a chronic cycle of repeated ulceration of the intestine followed by re-epithelialization with increasing production of aneuploid cells [31]. There are two major types of CRC, sporadic and colitis-associated CRC that are distinguished by their histologic presentation and the timing/sequence of cellular mutations (Figure 3A) [29,31]. CRC has also been classified into four consensus molecular subtypes (CMS) based on the genes and metabolic pathways associated with them. They are microsatellite instability (MSI) immune [CMS1], canonical [CMS2], metabolic [CMS3] and mesenchymal [CMS4] [1,32,33]. Sidedness has a key role in CRC (Figure 3B) with right-sided CRC (cecum, ascending colon, hepatic flexure) having overall a worse prognosis than left-sided CRC (splenic flexure, descending colon, sigmoid, rectosigmoid) due to right-sided tumors being more resistant to chemotherapy [33,34]. The major risk factors for developing CRC include, male gender, increased age, inflammatory bowel disease, smoking, alcohol, obesity, red meat, and family history [33]. Gut microbiota commonly found in CRC include *Fusobacterium nucleatum* and *Escherichia coli (pks+)* which were found to be positively correlated with production of biomarkers for damaged intestinal epithelium including, diamine oxide, D-lactate and lipopolysaccharides (LPS) [35].

For more information about gut microbiota and CRC, the reader is referred to the recent review article “Gut microbiota alterations are distinct for primary colorectal cancer and hepatocellular carcinoma” [41].

## 4. Metabolomics as a Potential New Way to Diagnose and Classify Cancer

The reprogramming of energy metabolism is a hallmark of cancer, notably, an increased dependence on aerobic glycolysis which is often referred to as the Warburg effect. This switch from oxidative phosphorylation to glycolysis allows cancer cells to meet the anabolic demands that result from their dysregulated cellular growth and altered differentiation. Ultimately, cancer cells develop a unique metabolic profile that distinguishes them from normal cells [42,43,44,45]. If this idea is pursued a bit further, one may hypothesize that different tumors arising in different tissue types may each have a cancer metabolite profile that is distinct from that of another tumor type. From our discussion thus far, it can be seen that at the molecular level, gene mutations are distinct and impact the phenotypes of these tumors. Therefore, genomic re-programming in each cancer type influences its metabolome.

Metabolomics analyses allow one to measure the end products of both host and microbiota metabolism thus affording a snapshot of the current biochemical status of the subject at the time of sample collection. It is a non-invasive procedure requiring only blood or urine collection which encourages greater patient compliance to have more frequent screenings for cancer [46]. In the following review we will examine some of the most current studies using LC-MS and GC-MS metabolomics platforms in order to compare their results and test our hypothesis that different tumors, notably, from EC, GC and CRC, have distinguishing serum metabolite profiles. The specific criteria used to select papers for this review are the following: (1) human studies, (2) serum/plasma studies only (with the view of minimizing result variability due to different sample type handling techniques and to exploring applications to the clinic), (3) LC-MS or GC-MS studies (to eliminate variable results due to use of different platforms) and (4) papers that were published within the past five years only. The studies selected were both untargeted and targeted metabolomics studies and this is indicated for each study. In papers where both discovery and validation cohorts were employed, results from the discovery set were used for our analysis.

## 5. The Metabolomics Profile of Gastric Cancer

In this section, we will begin by examining recent GC cancer metabolomics papers and extracting data regarding the most significantly different metabolites between GC and healthy controls found in blood. Table 1 summarizes three individual studies plus the findings of a meta-analysis for nine GC studies in blood [2,10,47,48]. Metabolomics studies have also been performed in urine [49] but as we are comparing three different cancers (GC, EC, CRC), only one sample type (blood) was used for the comparison. The main reason for choosing blood over urine samples is that 24 h urine samples need to be collected in order to get accurate metabolite detection and quantification. This is often inconvenient for the patient making complete compliance more difficult, whereas a fasting blood sample is much easier to obtain. The goal here was to examine potential metabolite profiles that could discriminate between the three cancers based on findings from recent studies. One study has also been performed to distinguish between blood metabolites found in patients with precancerous GC lesions and these are summarized in Table 2 [10].

The first study was a meta-analysis of nine plasma metabolomics studies published up to 2018 [47]. This study was counted as only one study because older data were embedded in the list of differential metabolites. Table 1 is a description of the patient cohorts used in all four studies.

Correa’s Cascade was pioneered by Pelayo Correa [50] and is a detailed histological classification of all of the histological changes that occur in the gastric epithelium that start with infection with *Helicobacter pylori*, proceeds to the formation of precancerous lesions and eventually culminates in GC. Table 3 above represents a summary of the differential metabolites from GC from an isolated recent study which looked at patients in various stages of Correa’s Cascade and did a metabolomics study on their serum [10]. Those which were deemed the most discriminatory were quantified and the ratio of the precancerous state/GC values are also listed in the table. Next, looking back at Table 2, the metabolites that were reported unanimously as well as those which were reported three out of four times as being differential between non-GC and GC patients are listed in Table 4 below. As these results were generated in different labs on different days and in some cases, different years, they can be considered as fairly reproducible although the number of studies is small. If one compares the metabolites in Table 4 with Table 3 one observes that four out of the five metabolites in Table 4, alanine, histidine, tryptophan and asparagine, show changes over the course of the precancerous stage of the disease relative to GC, This provides some evidence that these metabolites could be useful for early detection of precancerous stages of GC. If we then take the metabolites in Table 4 and enter them into the MetaboAnalyst program, one can find out the metabolic pathways that incorporate the listed metabolites. This information has been added to Table 4.

Another output from the MetaboAnalyst software is the impact map for the various metabolic pathways resulting from the pathway enrichment analysis (Figure 4). It can be seen that Phe, Tyr and Trp biosynthesis has been rated as important (indicated by dark red color), has several differential metabolites in the pathway (indicated by the size of the circle) and has the greatest impact (indicated by the *x*-axis).

The metaboloites listed in Table 4 are highyly relevant to cancer metabolism.It is well-known that oxidative phosphorylation of glucose is impaired in the mitochondria of cancer cells as a result of the Warburg effect and thus, the number of the acetyl-CoA molecules derived from glucose is significantly reduced. Instead, cancer cells rely on upregulation of amino acid biosynthesis and metabolism to replenish TCA cycle intermediates to generate ATP [2,51] Trp catabolism is important for the production of Acetyl-CoA for use in the TCA cycle and both His and Trp are used for the anabolism of one-carbon units for the synthesis of nucleotides for DNA and RNA biosynthesis [52]. Kynurenine, a major metabolite of Trp via the enzyme indoleamine-2,3-dioxygenase (IDO1), induces immunosuppression by binding to and activating the transcription factor aryl hydrocarbon receptor (AhR) [53,54]. This inhibits the ability of immune-tolerant dendritic cells (DCs) and regulatory T cells to target and eliminate cancer cells [55,56].

Glu is also used by cancer cells in the TCA cycle and it can be biosynthesized by the transamination of His and Arg and/or by the deamination of glutamine [51,57]. Thr catabolism via Thr dehydrogenase produces Gly and acetyl-CoA which can also feed the TCA cycle [58]. Met can also enter the Met-folate cycle to produce 1-carbon units for nucleobases [59]. All of the abovementioned amino acids were found to be reduced in the serum of GC patients possibly due to the upregulation of these metabolic pathways in the tumor. The Arg-Ornithine-polyamine pathway may be used to synthesize polyamines that promote the proliferation of cancer cells. Ornithine can be converted to citrulline in the urea cycle to replenish Arg supplies as Arg is decreased in most tumors [57,60,61]. Tyr is important for integration into proteins that activate important oncogenic signaling pathways such as Kras [62].

There is much interest in targeting amino acid metabolism to treat cancer. The alanine-serine-cysteine transporters 2 (ASCT2, encoded by SLC1A5 has been spotlighted as a therapeutic target because it is the primary glutamine transporter [57]. Gln is the amino acid that is consumed the most by cancer cells and inhibition of glutaminase which converts Gln to Glu, by CB-839 is in pre-clinical and clinical trials [63,64]. The enzyme glutamate dehydrogenase is responsible for the bioconversion of Glu to α-ketoglutamate for use in the TCA cycle and inhibition of this enzyme has been used to inhibit tumor growth [65]. Asparaginase, an injectable enzymatic drug that degrades asparagine in the plasma and is used as a treatment for acute lymphoblastic leukemia. The lack of Asn in cells cause apoptosis [66] Kinase inhibitors such as, imatinib and dasatinib, which act on receptors that bind Tyr containing proteins have been used successfully for treatment of GI cancer [67]. IDO1 inhibitors are actively being evaluated in clinical trials [59] The IDO1/Kynurinine/AhR pathway is also being investigated for its therapeutic potential [68]. Figure 4 is a diagram of the pathway impact for the discriminating metabolites just discussed.

In summary, although differences in study design and assay conditions resulted in many non-reproducible differential metabolites between the various studies there were still some which were reported consistently. Pathway enrichment analysis was performed on those metabolites and their significance to cancer and some therapeutic opportunities were discussed. In the next section we are going to use the same approach to examine the metabolomics of EC.

## 6. The Metabolomics Profile of Esophageal Cancer

Using the same technique used for GC we are now going to examine the recent serum metabolomics findings for esophageal cancer (EC). In the interest of saving space, the long table listing all of the differential metabolites for EC vs. non-EC patients has been moved to a Appendix A. Only the shorter tables listing those metabolites most consistently reported is given below in Table 5.

Using the same reasoning as for GC, Table 5 is a shortened list of differential metabolites based on reproducibility of reporting by different labs. There were four studies so as before we will allow reporting of a metabolite at least three times to count as a biomarker. Again, based on the results tabulated in Appendix A, there were many differences found in the results overall with some labs detecting many more changes in lipids. This disparity in results is reasonably due to the differences between labs in sample handling, chromatography (GC vs. HPLC) or detection techniques. However, once again there was good agreement between the labs for five differential metabolites as shown in Table 6. Both Tyr and Trp were detected for GC but for EC, three lipids, linoleic acid, oleic acid and palmitoleic acid were found three out of 4 studies.

One metabolic alteration in cancer is the accumulation of free fatty acids (FFAs) which enhances proliferation [73]. Fas-associated actor-1 (FAF1) contains a binding motif for unsaturated FAS but not for saturated FAs such as palmitic acid. FAF1 is a protein that facilitates the degradation of β-catenin [74]. Oleic acid and linoleic acid were both found to bind to FAF1 in various cancer cells lines and stabilize β-catenin, a transcriptional co-activator that stimulates expression of genes that drive cell proliferation [75]. Palmitoleic acid is an unsaturated FA which can also be biotransformed to palmitic acid which can then be used to produce more oleic acid [72] Palmitoleic acid can also be produced from palmitic acid by Δ9 desaturase [76]. Increased expression of β-catenin has been observed in ESCC and was correlated with a poor patient prognosis [77]. Elevated linoleic acid metabolism has been previously reported in EC patients [78]. Figure 5 is the pathway impact diagram that was generated from the pathway enrichment analysis done on the differential metabolites listed in Table 6.

From the Impact map generated by Metaboanalyst, one can see that the pathway with the highest impact rating is linoleic acid metabolism consistent with previous reports [78]. In the next section we are going to examine CRC in the same way as done for GC and ESCC.

## 7. The Metabolomics Profile of Colorectal Cancer

The last cancer we are going to use in our comparison analysis is CRC. As was done for EC, the long table listing all differential metabolites by each of the studies has been moved to the Appendix A. Appendix A is a listing from six different recent studies of the differential metabolites of CRC vs. non-CRC patients. The first reference listed was a meta-analysis of CRC studies published up to 2018 and the metabolites listed were found to be the most significant blood biomarkers (reported 5+ times) [47]. The meta-analysis was counted as a single study in our analysis because it incorporated some older papers. Table 7 summarizes the information provided for the participants in the six studies. As before, our sole purpose is to examine those metabolites that were measured and determined to be differential for cancer the most reproducibly.

As one see from Appendix A, the complete list of differential metabolites for CRC vs. non-CRC patients, there was not one metabolite that was unanimously detected by all of the studies. However, if we consider at least four out of six to be a possible biomarker, we will now examine those metabolites by pathway enrichment analysis for our final comparison to EC and GC cancers. Table 8 is a list of those metabolites reported by at least 4 out of 6 studies.

The metabolites which we have not been discussed and are distinct for CRC are Phe, Cys and lactic acid. Phe is important for the biosynthesis of Tyr whose importance to cancer cells has already been discussed extensively in the section related to GC [57]. Significant amount of ROS are generated in cancers due to their increased proliferation and oxidative stress and can cause cell death. Therefore, cancer cells must have increased antioxidant defenses to neutralize their increased ROS production. Glutathione is essential in maintaining redox balance in all subcellular compartments [59]. Production of glutathione requires Glu, Gly and Cys with Cys being the most critical component because of its thiol group. Inhibiting cysteine uptake has been shown to reduce cancer cell viability which was caused by uncontrolled oxidative stresses [83]. Cysteine can be imported into cells either directly or in its oxidized form, cystine, through the cystine/glutamate antiporter system xc− (xCT) [59]. Studies have looked at the efficacy of xCT inhibitors for cancer treatment [84,85]. Lactic acid is the end-product of glycolysis and would be expected to raise higher for cancer patients than HCs [57,76]. CRC Patients with higher serum Lactic acid levels were found to have poor prognosis, especially for metastatic CRCb [86]. Exogenous lactate derived from metabolism of lactic acid producing bacteria in the gut can serve as a fuel source for oxidative cancer cells and cause upregulation of monocarboxylate transporter 1 (MCT1). Therefore, lactate is considered as a tumor promoting metabolite as it can influence angiogenesis, amino acid metabolism, histone deacetlyases and immune escape contributing to cell migration [87]. Figure 6 is the pathway impact diagram generated by Metaboanalyst from the pathway enrichment analysis done on the differential metabolites listed in Table 8.

## 8. Comparative Analysis of the Metabolomes for GC, EC and CRC Along with Their Metabolic Pathways

We have finally reached a point where we can compare the altered metabolomes of EC, GC and CRC based on our data analysis of the current metabolomics literature. Table 9 indicates which altered metabolites are shared by the three cancers. 

Examination of the Table 7 reveals that both tryptophan and tyrosine are altered metabolites for all three of the cancers. Alanine perturbation is shared between GC and CRC but other than tryptophan, tyrosine and alanine, all other metabolites are distributed individually among the three cancers indicating a unique metabolome for each cancer. If you input the metabolites for a particular cancer into the MataboAnalyst software, an impact diagram is generated as shown in Figure 4, Figure 5 and Figure 6. Each filled circle corresponds to a metabolic pathway. The x-coordinate (Pathway Impact) indicates the extent of pathway influence by the perturbed metabolites. The point size is related to the diversity of the pathway, based on the number of metabolites contained in the pathway. The ordinate represents the negative logarithm of the *p*-value obtained from the enrichment analysis done by the MetaboAnalyst software. The *p*-value (measure of pathway importance) is encoded in the color of the circles with red being the highest *p*-value. One also is given a list of the matched metabolic pathways for each metabolite entered into the program. These pathways are compared between the three cancers in a heatmap (Figure 7). showing that many pathways are shared between the three cancers which may be due to the overlap of different metabolite involvement in the various pathways.

Although many metabolic pathways are shared there are a few which are assigned to only one cancer type. Shared pathways are important for designing therapeutics because a drug that targets a shared pathway can have an effect on more than one cancer type. On the other hand, targeting the unshared pathways could provide an opportunity for fewer side effects as the drug would be more selective. Unshared metabolites and pathways are more useful as biomarkers and diagnostic tools than those which are shared. However, going back to Figure 4, Figure 5 and Figure 6, the impact of the shared pathway, Phenylalanine, tyrosine, tryptophan biosynthesis ranks as EC < GC < CRC and thus although shared, targeting this pathway may have different effects on different cancer types. In contrast, the tyrosine and tryptophan metabolism pathways appear to have an equal impact on all three cancers.

Figure 8 is an example of another type of data output from the MetaboAnalyst program. Every pathway which is mapped as a circle on the Impact diagram can be mouse clicked and the pathway will be displayed. Mouse clicking on each square of the pathway gives you the name of the compound as shown in Figure 8 where these have been typed in separately. Figure 8 also shows some of the metabolic intermediates of tryptophan in bold red. These are metabolites that have been reported as differential by the studies presented in this review article at least once. The multiple branches in the diagram also allows one to hypothesize that one could target one branch of a pathway and not completely inhibit the functions of a metabolite such as tryptophan. An example would be inhibition of formylkynurenine to block production of kynurenine and allowing synthesis of serotonin, an important neurotransmitter to occur (Figure 8). It has been previously reported that high enzyme activity of the kynurenine pathway is associated with immune escape, tumor progression and migration [87]. It has also been demonstrated that kynurenine inhibits T-cell proliferation and induces T-cell apoptosis, leading to immune tolerance and a tumor progression/metastatic microenvironment [88].

Although there has been inconsistency in the detection of tryptophan metabolites, at least one attempt has been made to develop a targeted metabolomics assay for tryptophan and some of its metabolites [89]. A follow-up study utilizing this assay was done and five metabolites (i.e., tryptophan (TRP), kynurenine (KYN), 5-hydroxytryptophan (5HTP), 5-hydroxyindole-3-acetic acid (5HIAA), 5-hydroxytryptamine (5HT)) were quantified and compared between healthy controls, ESCC and metastatic ESCC patients. Their results showed that the ratios of KYN/TRP, 5HTP/TRP, 5HIAA/TRP and 5HT/TRP exhibited a similar up-regulated tendency among healthy controls, ESCC and mESCC patients and that the ratios of KYN/TRP and 5HTP/TRP were significantly different between metastatic ESCC and ESCC patients [90].

## 9. The Role of Gut Microbiota Produced Metabolites in GI Cancer

Increases in certain microbial metabolites also play a role in GI cancer. As we have just pointed out for CRC, increased amounts of lactate derived from lactic acid producing bacteria are able to cause upregulation of the MCT1 transporter to allow increased amounts of lactate into the cancer cell where it acts a tumor promoting substance [91]. Lactate-derived pyruvate can stabilize hypoxia-inducible factor-1 (HIF-1) by inhibiting HIF polyhydroxylases which in tumor endothelial cells, stimulates angiogeneisis [92]. Lactate also mediates M2-like polarization of tumor associated macrophages contributing to immunosuppression and immune escape of cancer cells [93].

While butyrate has been shown to have beneficial effects, in the context of APC mutations as is found in nearly all of CRC, butyrate was shown to promote proliferation of aberrant epithelial cells contributing to increased cancer polyp formation [94]. Acetaldehyde, especially in saliva is mainly of microbial source and contributes to upper GI cancer by causing molecular damage and mutagenesis [95]. The esophageal microbiome is a reflection of the oral microbiome [96] and oral pathogens have been implicated in CRC as well [97].

Secondary and tertiary bile acids (BAs) are bacterial metabolites that have been implicated in GI carcinogenesis [98]. In a recent study of human BA reflux gastritis, it was found that in patients with BA reflux gastritis that there were higher amounts of conjugated primary and secondary BAs, notably, glycocholic acid (GCA), glycochenodeoxycholic acid (GCDCA), glycodeoxycholic acid (GDCA), taurodeoxycholic acid (TDCA), taurocholic acid (TCA) and taurochenodeoxycholic acid (TCDCA) in their gastric juice In contrast, normal and gastritis patients without BA reflux had equal amounts of conjugated and unconjugated BAs in their gastric juice [99].

## 10. Summary and Perspectives

In this review we sought those biomarkers that were present in most of the studies. If a differential metabolite was reported ≥70% of the time by different labs in different parts of the world in different years, then we felt this could not be a coincidence. All of the biomarkers which were selected by our technique were highly relevant to cancer metabolism which was highlighted in a discussion following each cancer section of the review. Metabolomics shows promise for cancer screening but a more cooperative effort between labs must be established to standardize assays and study designs for this technique to be used in the clinic such that on any given day in any part of the world a patient will receive an accurate assessment of his health. The fact that very different types of metabolites were detected by different labs with relatively low reproducibility is a good indication that assay conditions were very different for each of the labs. While differences in technique and study design can explain differences in results, a true biomarker should be directly keyed to the disease. In other words, if the tumor type is present than the differential biomarker should be there. Only if protocols become uniform can this type of biomarker be truly discovered. Independent labs who use their time and resources to repeat and validate an assay should perhaps be given publication opportunities as this is important to communicate to the metabolomics community. Thus, the publication industry has a big role to play as publication should be used to disseminate scientific findings that are not just unique but also for the benefit of the public, in this case, cancer patients. Allowing other scientists to read about good result reproducibility will encourage the use of those assays and produce more consistent findings that will eventually lead to a trustworthy clinical assay. Funding agencies should also provide help to those labs which are willing to perform assay validation work. Once consistent biomarkers are found, there is much more basic science work to be done in the metabolomics field to develop more targeted assays to provide information about cancer vulnerabilities and enable a more selective targeting of a metabolic pathway while preserving as much normal physiological function as possible. During this process we shall also learn more about why established drugs work better for some cancers. The main point being that the assays must eventually become standardized and more uniform. If the ideas put forth in this last paragraph are seriously considered as worthwhile, we believe that having metabolomics assays implemented in the clinical lab is still very winnable.

## 11. Conclusions

In conclusion, serum metabolomics has a real potential for use as a cancer-screening tool. Cancer tumors are distinguishable between different types of cancers based on histology, genomic backgrounds and mutations, and also, as found in this review, their metabolic phenotypes. The changes in metabolism are due to cancer cell reprogramming which enables higher output of biomolecular building blocks necessary for increased proliferation and ultimately, survival in terms of immune escape and the ability to metastasize. In addition, the metabolic output of disease-induced composition changes in the gut microbiota play a role in the cancer metabolic phenotype. Serum metabolomics is a way to identify cancer related changes due to both patient and gut mictobiota metabolic output. A simple blood sample from a patent, coupled with trustworthy, uniform measurement procedures may someday allow earlier cancer detection and better patient outcomes. 

## Figures and Tables

**Figure 1 cancers-13-00720-f001:**
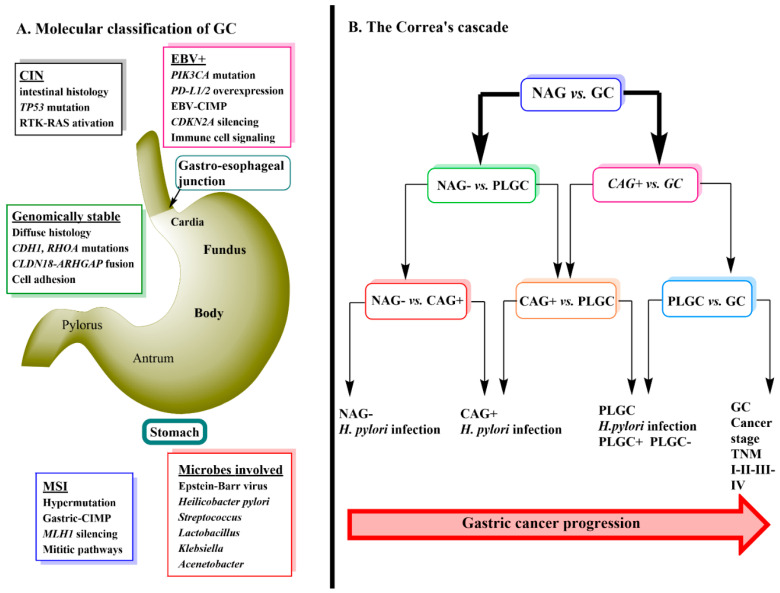
(**A**) Molecular classification of gastric cancer. Four tumor subgroups have been identified: positive for Epstein-Barr virus (EBV, 9%), microsatellite unstable (MSI, 22%), genomically stable (20%) and chromosomally unstable tumors (CIN, 50%). Genomically stable tumors are ~73% correlated histologically with diffuse tumors. CIN tumors are mainly gastro-esophageal junction adenocarcinomas (intestinal type tumor) and EBV tumors are found predominantly in either the fundus or body of the stomach with 81% of these found in males. Women have more MSI tumors (56%, intestinal, antrum location) [12]. CIN is characterized by DNA aneuploidy, translocation of chromosomes and mutations in proto-oncogenes and tumor suppresser genes [13] while MSI tumors are characterized by DNA mismatch repair defects caused by epigenetic events such as hypermethylation [14]. (**B**) Correa’s cascade. The progression of GC occurs via multiple precancerous lesions and is often described by Correa’s cascade. The stepwise progression starts with superficial non-atrophic gastritis (NAG) which can then advance to precursor lesions of GC (PCGL) which proceeds through multifocal atrophic gastritis, intestinal metaplasia and dysplasia before it ultimately becomes GC. *H. pylori* infection occurs at all stages in the cascade. Abbreviations: cadherin-1 (CDH1), claudin18 (CLDN18), CpG island methylator phenotype (CIMP), cyclin-dependent kinase inhibitor 2A (CDKN2A, tumor suppressor), MUTL homolog 1 (MLH1, tumor suppressor), programmed death ligand 1/2 (PD-L1/2, proto-oncogene), Phosphatidylinositol-4,5-Bisphosphate 3-Kinase Catalytic Subunit Alpha (PI3KCA, proto-oncogene), ras homolog gene family member A (RHOA, proto-oncogene), rat sarcoma virus (RAS, proto-oncogene), receptor tyrosine kinase (RTK), Rho GTPase activating protein (ARHGAP), tumor protein 54 (TP53, tumor suppressor) Reference for microbiota involvement [15].

**Figure 2 cancers-13-00720-f002:**
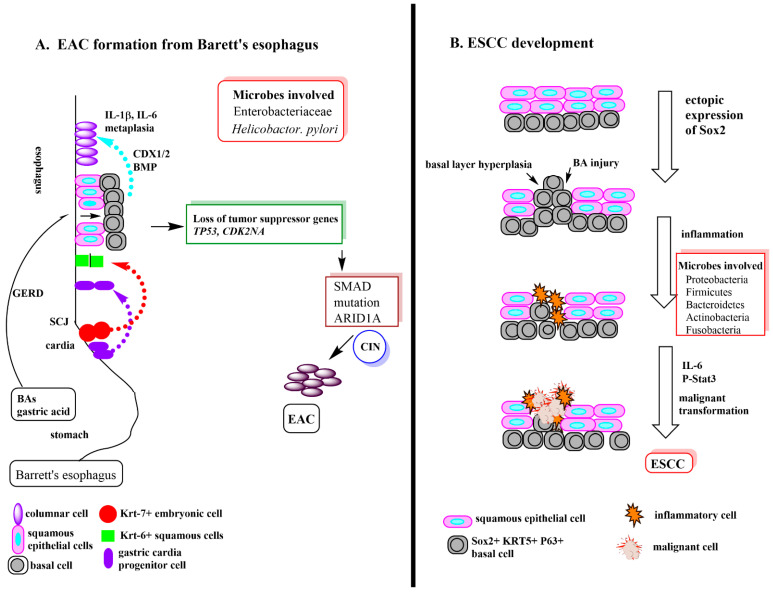
(**A**) The formation of Barrett’s esophagus and transition to esophageal adenocarcinoma. There are several popular theories as to how Barrett’s esophagus is formed. The three most current are pictured diagrammatically in this figure. (1) The purple arrow indicates the migration of gastric cardia progenitor cells to the esophagus due to inflammation driven expansion of these cells. (2) The red arrow indicates the inflammation driven migration of Krt-7+ squamous epithelial cells at the SCJ junction to replace the normal Krt-6+ in the esophagus. (3) The blue arrow indicates what happens after GERD damage opens up squamous cell junctions and allow acid damage of the basal cells. In response to damage the basal cells express CDX1 and BMP to cause the basal cells to transform into metaplastic columnar cells along with further increased damage due to inflammation. Once Barrett’s esophagus is established, further inflammation induced damage of the columnar cells leads to cell mutations, notably, loss of tumor suppressor genes, *TP53* and *CDK2NA* which in turn, increases mutation of SMAD (leads to dysregulated cell growth) and the expression of AT-rich interactive domain-containing protein 1A, ARID1A) (part of a protein complex responsible for activating genes normally silenced by chromatin structure). The final result is chromosome instability (CIN) and malignant transformation to EAC [19]. Abbreviations: AT-rich interactive domain-containing protein 1A (ARID1A), bile acid (BA), bone morphogenetic protein (BMP, growth factor), caudal homeobox gene 1 (CDX1), chromosomal instability (CIN), cyclin-dependent kinase inhibitor 2A (CDKN2A, tumor suppressor), esophageal adenocarcinoma (EAC), gastric acid esophageal reflux disease (GERD), interleukin (IL) (IL-1β, IL-6, inflammatory cytokines), squamocolumnar junction (SCJ), tumor protein 54 (TP53, tumor suppressor) (**B**) ESCC development. The development of esophageal squamous cell carcinoma (ESCC) begins with the ectopic expression of Sox2 in basal cells. Sox2 is critical for maintaining self-renewal and appropriate proportion of basal cells in adult tracheal epithelium. However, its overexpression gives rise to extensive epithelial hyperplasia. BA injury to basal cells leads to more basal cell hyperplasia and inflammation. Increased levels of the inflammatory cytokine, IL-6, and increased levels of activated P-Stat3 results in malignant transformation to ESCC. Abbreviations: Signal transducer and activator of transcription 3 (STAT3), SRY (sex determining region Y)-box 2 (Sox2) Reference for microbiota involvement [15].

**Figure 3 cancers-13-00720-f003:**
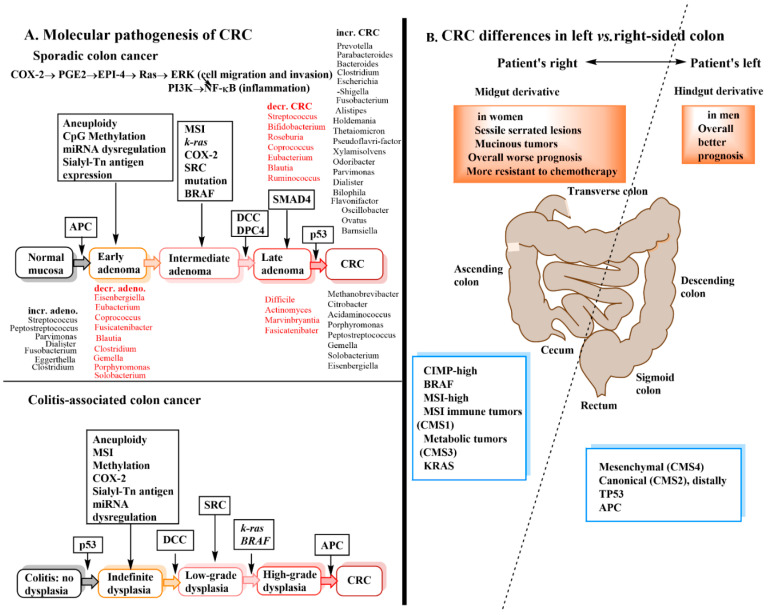
(**A**) The molecular pathogenesis of CRC. There are two broad classifications of CRC based on differences in the sequence of mutations leading to the cancer as well as their histological presentation. Sporadic CRC (SCRC) (top) develops from dysplasia of 1 or 2 discrete foci of the colon. The primary event in the formation of early adenomas is the loss of APC protein function. The APC protein is a negative regulator of beta-catenin that helps to control how often a cell divides and is the product of the tumor suppressor gene APC. Early adenoma is also accompanied by expression of the mucin-associated carbohydrate antigen, sialyl-Tn [36], miRNA dysregulation dysregulation and progression to an intermediate stage of adenoma that is characterized by activation of the oncogenic *k-ras* gene and COX-2. COX-2 induction leads to production of the pro-inflammatory prostaglandin PGE_2_ which in turn binds to EP1-4 leading to several important pathways (shown in the figure) that promote CRC progression from early to intermediate adenoma stages [37]. The intermediate adenoma stage is also characterized by SRC overexpression and subsequent promotion of CRC cell proliferation and survival [38]. The loss of the tumor suppressor genes *DCC* and *DPC4* often occur in the progression from intermediate to late adenoma [39]. The advancement to CRC from late adenoma is characterized by loss of p53 as the final mutation event. Colitis-associated CRC (CCRC) first presents as dysplastic lesions that are polyploidy, flat, localized or multifocal and which are the product of chronic inflammation. Rather than distinct polyps (adenomas) there is instead a large spreading region that often indicates removal of the entire colon and rectum [40]. There are some major differences in the timing and sequence of mutations from those seen for SCRC. Rather than adenoma stages there are instead varying degrees of epithelial dysplasia in the staging of the disease. Abbreviations: sporadic CRC (SCRC), colitis-associated CRC (CCRC), adenomatous polyposis coli (APC), microsatellite instability (MSI), Kirsten rat sarcoma viral oncogene homolog (k-ras), cyclooxygenase-2 (COX-2), prostaglandin E_2_ (PGE2), dependence receptor in colorectal cancer (DCC), deleted in pancreatic cancer-4 (DPC4), proto-oncogene tyrosine protein-kinase Src 5 (SRC). (**B**) CRC differences in left vs. right-sided colon. The molecular features of left-sided (distal) colon cancers are different from right-sided (proximal) colon cancers. Based on the different gene expressions in CRC, four consensus molecular subtypes (CMS, 1-4) have been identified; MSI immune (CMS1), canonical (CMS2), metabolic (CMS3), and mesenchymal (CMS4). The left-sided cancers have better prognosis because they are less resistant to chemotherapy (CMS2, 4). Sidedness is particularly relevant during metastasis and is predictive of drug response [34].

**Figure 4 cancers-13-00720-f004:**
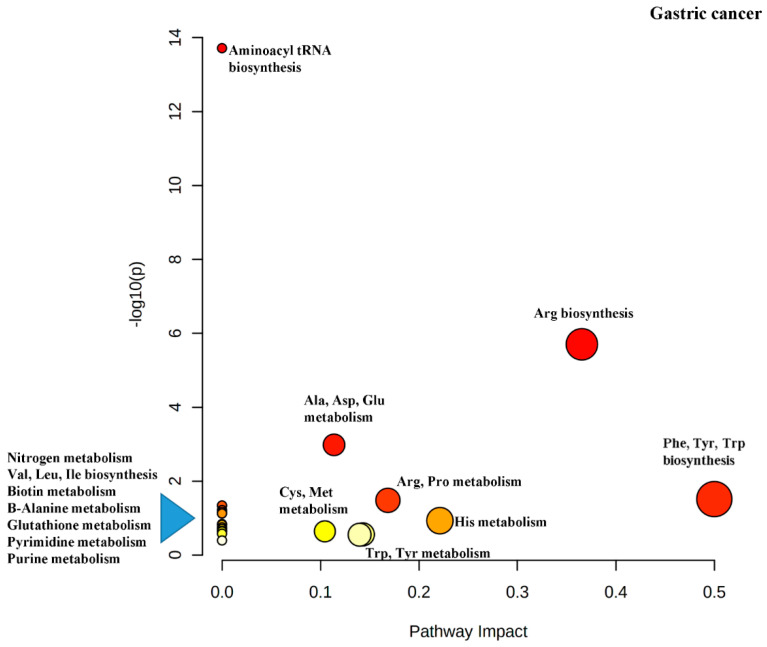
Pathway analysis performed using the altered metabolites from metabolomics data for gastric cancer. Each filled circle corresponds to a metabolic pathway. The x-coordinate (Pathway Impact) indicates the extent of pathway influence by the perturbed metabolites. The point size is related to the diversity of the pathway, based on the number of metabolites contained in the pathway. The ordinate represents the negative logarithm of the *p*-value obtained from the enrichment analysis done by the MetaboAnalyst software. The *p*-value (measure of pathway importance) is encoded in the color of the circles with red being the highest *p*-value.

**Figure 5 cancers-13-00720-f005:**
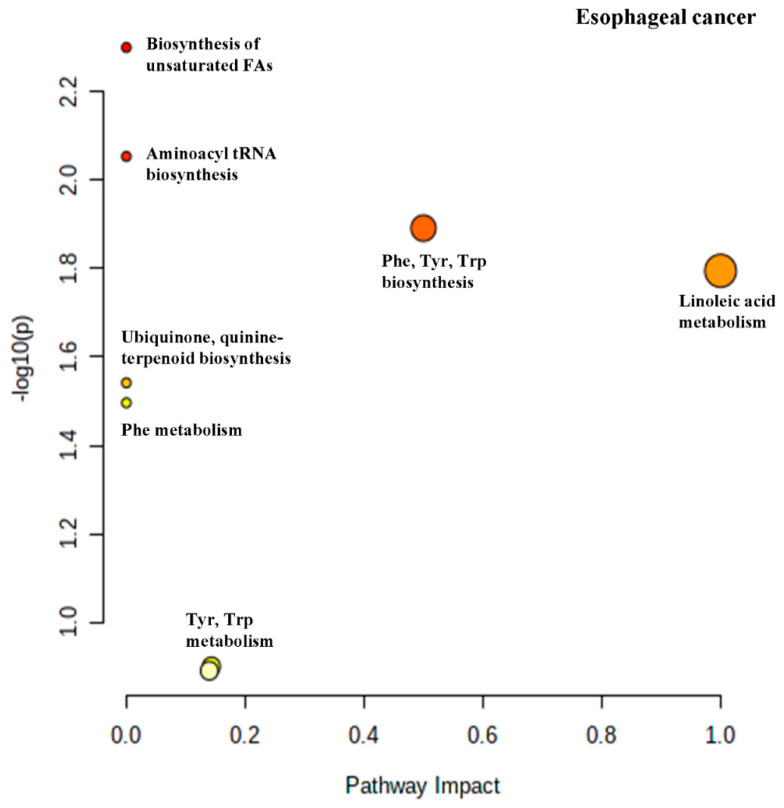
Pathway analysis performed using the altered metabolites from metabolomics data for esophageal cancer (EC) (refer to the legend of Figure 4 for the diagram explanation).

**Figure 6 cancers-13-00720-f006:**
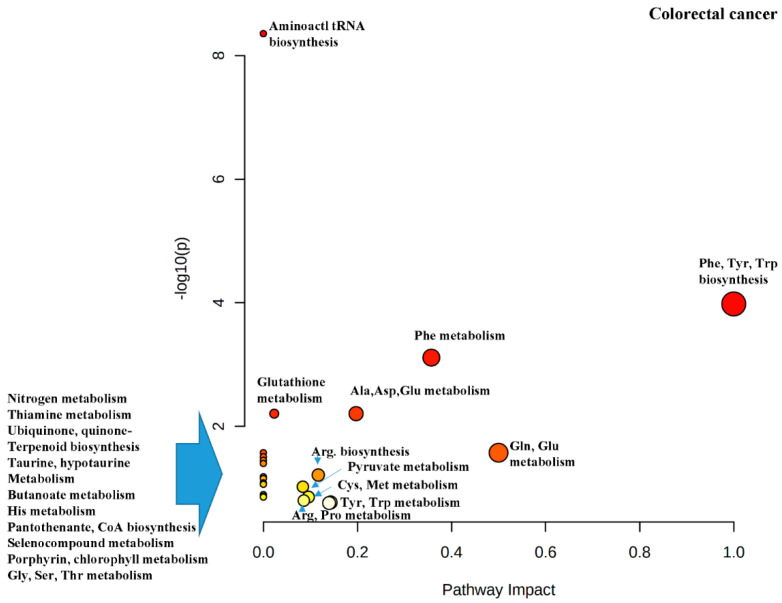
Pathway analysis performed using the altered metabolites from metabolomics data for colorectal cancer (CRC) (refer to the legend of Figure 4 for the diagram explanation).

**Figure 7 cancers-13-00720-f007:**
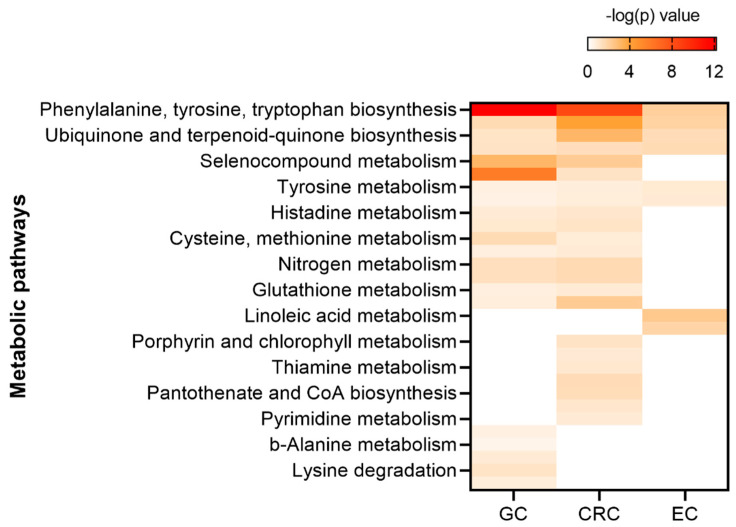
Comparison of the altered metabolic pathways for GC, EC and CRC. The metabolites demonstrated in Table 9 were uploaded to map to KEGG metabolic pathways for over-representation analysis and pathway topology analysis. −log(p) value represents the relevance and weight.

**Figure 8 cancers-13-00720-f008:**
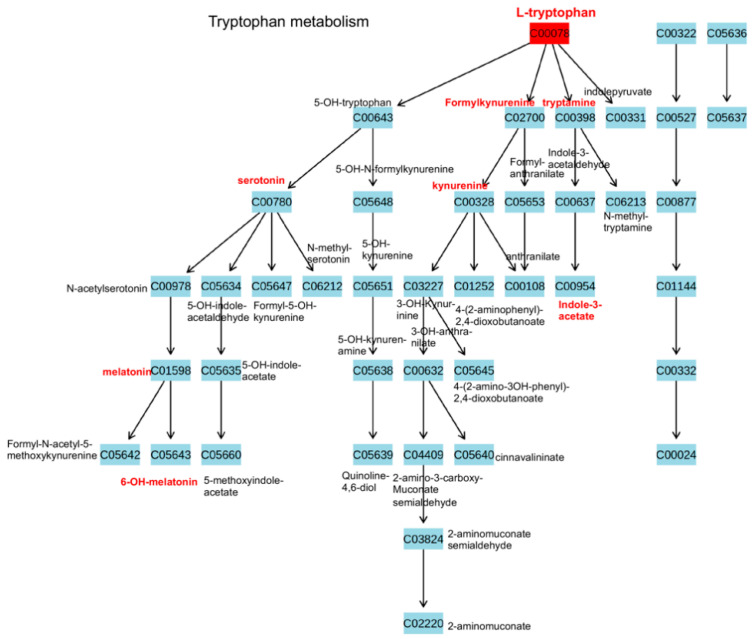
Tryptophan metabolism. The metabolites highlighted in red lettering have been reported at least once in the studies presented in this review. Tryptophan was found to be altered in all three cancers and may be a candidate for the development of a targeted metabolomics assy.

**Table 1 cancers-13-00720-t001:** Description of patient cohorts and study types used in GC studies.

Reference [46]	Reference [10]	Reference [2]	Reference [48]
184 GC/208 HC Unmatched Case control study Untargeted metabolomics	20 GC/19 HC Unmatched 68/43 GC/HC mean age 8 F/12 M (GC) Targeted metabolomics panel of 216 metabolites	84 GC/82 non-GC Unmatched 28–79 age GC 25–82 age non-GC 45 M/39 F (GC) Targeted metabolomics panel of Amino acids	104 GC/50 HC Unmatched Untargeted metabolomics

**Table 2 cancers-13-00720-t002:** Differential metabolites (GC vs. non-GC) for gastric cancer.

Metabolite [47] (Meta-Analysis *n* = 9)	Metabolite [10]	Metabolite [2]	Metabolite [48] (Xiu)
		Glycine ↑	Glycine ↓
Tyrosine		Tyrosine ↓	Tyrosine ↓
Phenylalanine			Phenylalanine ↓
Alanine	Alanine ↓	Alanine ↑	Alanine ↓
Threonine	Threonine		Threonine ↓
Isoleucine			Isoleucine ↓
Histidine	Histidine ↓	Histidine ↓	Histidine ↓
Taurine	Taurine		
Arginine		Arginine ↓	Arginine ↑
Leucine			Leucine ↓
Methionine	Methionine		Methionine ↓
Valine			Valine ↑
		Serine ↑	Serine ↑
Tryptophan	Tryptophan ↓	Tryptophan ↓	Tryptophan ↓
Fumarate			
			Cystine ↓
Asparagine	Asparagine ↑	Asparagine ↑	Asparagine ↓
Lysine	Lysine	Lysine ↑	Lysine ↓
Propanoic acid			
Pyruvic acid			
			Glutamate ↓
Glutamine		Glutamine ↓	Glutamine ↓
Citrulline	Citrulline	Citrulline ↓	
Spermidine	Spermidine ↑		
3-Hydroxypropionic acid			
Metabolite [47]	Metabolite [10]	Metabolite [2]	Metabolite [48] (Xiu)
Anthranilic acid			
Ornithine	Ornithine ↑	Ornithine ↑	
Sarcosine	Sarcosine		
Creatinine			
2-Hydroxybutyrate			
3-Hydroxyisobutyric acid			
	Erythro-isoleucine ↑		
	Symmetric dimethylarginine ↑		
	hydroxytetradecadienylcarnitine↑		
	Methionine sulfoxide ↑		
	Tetradecanoylcarnitine		
	Hexadecadienylcarnitine ↑		
	Octadecanoylcarnitine ↓		
Xanthurenic acid	Xanthurenic acid		
	Phenylacetylglutamine ↑		
	Octadecenoylcarnitine		
	N-formylkynurenine		
Uric acid			
d-Glucose			
Melatonin			
Serotonin			
2-aminobenzoic acid			
l-Kynurenine			
Kynurenic acid			
Tryptamine			
3-Indoleactamide			
(Indol-3-yl)acetamide			
Indolacetic acid			
6-Hydroxymelatonin			
5-Methoxytryptamine			
Indolelactic acid			
Tryptophanol			
Propionic acid			
Quinolinic acid			
Niacinamide			
		Homocysteine ↓	

The arrows indicate reported increasing or decreasing trends relative to non-GC controls.

**Table 3 cancers-13-00720-t003:** The metabolite profile of GC across Correa’s cascade [10].

NAG/GC Ratio	CAG/GC Ratio	PLGC/GC Ratio
Alanine 1.32 ↑	Alanine 1.36 ↑	Alanine 1.39 ↑
Asparagine 1.12 ↑	Asparagine 1.08 ↑	Asparagine 1.10 ↑
Histidine 1.22 ↑ *	Histidine 1.18 ↑ *	Histidine 1.14 ↑ *
Erythro-isoleucine 1.03 ↑	Erythro-isoleucine 0.96 ↓	Erythro-isoleucine 0.96↓
Ornithine 0.91↓	Ornithine 0.82 ↓	Ornithine 0.64↓
Symmetric dimethylarginine 0.79 ↓	Symmetric dimethylarginine 0.84 ↓	Symmetric dimethylarginine 0.70↓ *
Hydroxytetradecadienylcarnitine 0.87 ↓	Hydroxyteradecadienylcarnitine 0.82↓ *	Hydrocytetradecadienylcarnitine 0.85↓ *
Methionine sulfoxide 0.93 ↓	Methionine sulfoxide 0.83↓ *	Methionine sulfoxide 0.81 ↓ *
Sarcosine		
Spermidine 0.92 ↓	Spermidine 0.88 ↓	Spermidine 0.94 ↓
Tetradecanoylcarnitine		
Hexadecanoylcarnitine 0.88 ↓	Hexadecanoylcarnitine 0.85 ↓	Hexadecanoylcarnitine 0.88 ↓
Octadecanoylcarnitine 1.33 ↑	Octadecanoylcarnitine 1.50 ↑ *	Octadecanoylcarnitine 1.50 ↑ *
Xanthurenic acid		
Phenylacetylglutamine 0.33 ↓ *	Phenylacetylglutamine 0.45 ↓*	Phenylacetylglutamine 0.45↓*
Tryptophan 1.63 ↑ *	Tryptophan 1.36 ↑ *	Tryptophan 1.31 ↑ *
	Lysine	
	Methionine	
	Threonine	
	N-acetylornithine	
	Hydroxyhexadecanoylcarnitine	
	Octadecenoylcarnitine	
	N-formylkynurenine	
		Taurine

* Indicates statistical significance (*p* < 0.05). The arrows indicate the trend relative to GC.

**Table 4 cancers-13-00720-t004:** Metabolites reported by all studies and three out of four studies listed in Table 2 and their associated metabolic pathways.

Metabolite [47] (Meta-Analysis *n* = 9	Metabolite [10]	Metabolite [2]	Metabolite [48] (Xiu)	Matched Pathways
Alanine	Alanine ↓	Alanine ↑	Alanine ↓	Aminoacyl-tRNA biosynthesis Alanine, aspartate, glutamate metabolism Selenocompound metabolism
Histidine	Histidine ↓	Histidine ↓	Histidine ↓	Aminoacyl-tRNA biosynthesis Histadine metabolism Β-alanine metabolism
Tryptophan	Tryptophan ↓	Tryptophan ↓	Tryptophan ↓	Aminoacyl-tRNA biosynthesis Tryptophan metabolism
Asparagine	Asparagine ↑	Asparagine ↑	Asparagine ↓	Aminoacyl-tRNA biosynthesis Alanine, aspartate, glutamate metabolism
Lysine	Lysine	Lysine ↑	Lysine ↓	Aminoacyl-tRNA biosynthesis Biotin metabolism Lysine degradation
Tyrosine		Tyrosine ↓	Tyrosine ↓	Aminoacyl-tRNA biosynthesis Phenylalanine, tyrosine, tryptophan biosynthesis Ubiquinone and terpenoid-quinone biosynthesis Phenylalanine metabolism Tyrosine metabolism
Threonine	Threonine		Threonine ↓	Aminoacyl-tRNA biosynthesis Valine, leucine, isoleucine biosynthesis Glycine, serine, threonine metabolism
Arginine	Arginine ↑		Arginine ↓	Aminoacyl-tRNA biosynthesis Arginine biosynthesis Arginine and proline metabolism
Methionine	Methionine		Methionine ↓	Aminoacyl-tRNA biosynthesis Cysteine, methionine metabolism
Glutamine		Glutamine ↓	Glutamine ↓	Aminoacyl-tRNA biosynthesis Arginine biosynthesis Alanine, aspartate, glutamate metabolism Glutamine and glutamate metabolism Nitrogen metabolism Glyoxylate and dicarboxylate metabolism Pyrimidine metabolism Purine metabolism
Citrulline	Citrulline	Citrulline		Arginine biosynthesis
Ornithine	Ornithine	Ornithine		Arginine biosynthesis Arginine and proline metabolism Glutathione metabolism

The arrows indicate reported increasing or decreasing trends relative to non-GC controls.

**Table 5 cancers-13-00720-t005:** Description of patient cohorts and study types used in EC studies.

Reference [69]	Reference [70]	Reference [71]	Reference [72]
24 EC/21 HC Matched 19 M/5 F EC 47 M/4 F HC 48–86 age EC 45–86 age HC Untargeted GC/MS	80 EC/80 HC Unmatched 53 M/27 F EC 45 M/35 F HC 59 EC/51 HC Mean age Untargeted	Two discovery phase cohorts 30 EC/30 HC x 2 Matched 63 EC/63 HC mean age 90% M for both EC and HC Untargeted	77 EC/84 HC 40–69 age range All subjects No gender info given Untargeted

**Table 6 cancers-13-00720-t006:** Metabolites reported by at least three out of four studies listed in Appendix A and their associated metabolic pathways.

Metabolite [69]	Metabolite [70]	Metabolite [71]	Metabolite [72]	Matched Pathways
Tryptopha n ↓	Tryptophan ↓	Tryptophan ↓	Tryptophan ↓	Aminoacyl-tRNA biosynthesis Tryptophan metabolism
Tyrosine ↑	Tyrosine ↓	Tyrosine ↓	Tyrosine ↓	Aminoacyl-tRNA biosynthesis Phenylalanine, tyrosine and tryptophan biosynthesis Ubiquinone and other terpenoid-quinone biosynthesis Phenylalanine metabolism Tyrosine metabolism
Linoleic acid	Linoleic acid		Linoleic acid	Biosynthesis of unsaturated fatty acids Linoleic acid metabolism
Oleic acid ↑		Oleic acid↑	Oleic acid ↑	Biosynthesis of unsaturated fatty acids
	Palmitoleic acid ↑	Palmitoleic acid ↑	Palmitoleic acid ↑	No pathways matched in MetaboAnalyst

The arrows indicate reported increasing or decreasing trends relative to non-GC controls.

**Table 7 cancers-13-00720-t007:** Description of patient cohorts and study types used in CRC studies.

Reference [47]	Reference [79]	Reference [80]	Reference [81]	Reference [82]	Reference [76]
1870 CRC 1857 HC Mixture of matched and unmatched gender not specified for all studies Ages not given Case control with Some nested-case studies Untargeted	22 CRC/15 M/7 F 45 HC/31 M/14 F Age and sex Matched Age range 49–84 Untargeted	320 CRC 201 M CRC 119 F CRC 148 M HC 106 F HC Age/sex matched Mean age 66 CRC 62 HC Untargeted	30 CRC 18 M/12 F Mean age 54 30 HC 18 M/12 F Mean age 55 Matched Untargeted	56 CRC 28 M/28 F Mean age 70 60 HC 30 M/30 F Mean age 68 Age/sex matched Untargeted	282 CRC 170 M/112 F Mean age 67 291 HC 178 M/113 F Mean age 67 Age/sex matched Unknown if targeted or untargeted

**Table 8 cancers-13-00720-t008:** Metabolites reported by at least four out of six studies listed in Appendix A and their associated metabolic pathways.

Metabolite [47]	Metabolite [79]	Metabolite [80]	Metabolite [81]	Metabolite [82]	Metabolite [76]	Matched Pathways
Glutamic acid			Glutamic acid	Glutamic acid	Glutamic acid	Aminoacyl-tRNA biosynthesis Glutathione metabolism Ala, Asp, Glu metabolism Nitrogen metabolism Gln, Glu metabolism Arginine biosynthesis Butanoate metabolism Histidine metabolism Porphyrin and chlorophyll metabolism Glyoxylate, dicarboxylate metabolism Arginine and proline metabolism
Phenylalanine		Phenylalanine		Phenylalanine	Phenylalanine	Aminoacyl-tRNA biosynthesis Phe, Tyr, Trp biosynthesis Phe metabolism
Alanine	Alanine		Alanine	Alanine	Alanine	Aminoacyl-tRNA biosynthesis Ala, Asp, Glu metabolism Selenocompound metabolism
Lactic acid	Lactic acid	Lactic acid	Lactic acid		Lactic acid	Pyruvate metabolism
Cysteine		Cysteine		Cysteine	Cysteine	Aminoacyl-tRNA biosynthesis Glutathione metabolism Thiamine metabolism Taurine, hypotaurine metabolism Pantothenate and CoA biosynthesis Glycine, Ser Thr Metabolism Cysteine and methionine metabolism
Tyrosine		Tyrosine		Tyrosine	Tyrosine	Aminoacyl-tRNA biosynthesis Phe, Tyr, Trp biosynthesis Phe Metabolism Ubiquinone, terpenoid-quinone biosynthesis Tyrosine metabolism
Tryptophan	Tryptophan			Tryptophan	Tryptophan	Aminoacyl-tRNA biosynthesis Tryptophan metabolism

**Table 9 cancers-13-00720-t009:** Comparison of the three cancers, GC, EC and CRC based on their respective altered metabolites.

GC	EC	CRC
Alanine		Alanine
Histidine		
Tryptophan	Trptophan	Tryptophan

Asparagine		
Lysine		
Tyrosine	Tyrosine	Tyrosine
Arginine		
Methionine		
Glutamine		
Citrulline		
Ornithine		
	Linoleic acid	
	Oleic acid	
	Palmitoleic acid	
		Phenylalanine
		Lactic acid
		Cysteine
		Glutamic acid

## Data Availability

The data presented in this study are available in this article (and Appendix A).

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
