# Peer review of "The Distinctive Serum Metabolomes of Gastric, Esophageal and Colorectal Cancers"

_cancers, 2021, doi:10.3390/cancers13040720_

Round 1

Reviewer 1 Report

The authors performed a review of recent papers that investigated metabolomics in 3 different cancers. These cancers, as elucidated in the manuscript, are leading causes of death and thus research into these cancers is important to further the underlying biology.
Overall, the paper sums up and presents the data retrieved from the different papers in an understandable if uninspired way. The reader understands that different metabolomic pathways contribute towards the different cancers. However, the presentation of the results is suboptimal and repetitive. Figures, such as heatmaps indicating absence/presence/direction of the metabolites, instead of tables would be much easier to understand and would be consuming less space.
The manuscript text needs considerable editing and correcting of the english used. Sometimes words are missing (e.g. "acid" in line 96) or the use of commas is wrong.
Unfortunately, something that is only alluded to in the manuscript is the different assays used per laboratory. This makes a comparison of the different papers, and thus the results, presented here quite challenging. At the very least a summary indicating if the data came from a targeted or untargeted approach would be needed.

Author Response

Point 1: The authors performed a review of recent papers that investigated metabolomics in 3 different cancers. These cancers, as elucidated in the manuscript, are leading causes of death and thus research into these cancers is important to further the underlying biology.
Overall, the paper sums up and presents the data retrieved from the different papers in an understandable if uninspired way. The reader understands that different metabolomic pathways contribute towards the different cancers. However, the presentation of the results is suboptimal and repetitive. Figures, such as heatmaps indicating absence/presence/direction of the metabolites, instead of tables would be much easier to understand and would be consuming less space.

Response 1: Two of the longer tables, 4 and 6 have been cut from the manuscript and are in an SI file as Table S1 and S2, respectively.  A heatmap of the last longer Table in the manuscript has been constructed and is Figure 7.

Point 2: The manuscript text needs considerable editing and correcting of the english used. Sometimes words are missing (e.g. "acid" in line 96) or the use of commas is wrong.

Response 2: The English was rechecked by a native English speaker and the word “acid” was inserted after deoxycholic in line 96.

Point 3: Unfortunately, something that is only alluded to in the manuscript is the different assays used per laboratory. This makes a comparison of the different papers, and thus the results, presented here quite challenging. At the very least a summary indicating if the data came from a targeted or untargeted approach would be needed.

Response 3:Thank you very much for taking time to read and assess our manuscript.  You pointed us towards more needed discussion for this manuscript.

We have inserted the following comments at the end of the Introduction (lines 219-221). “The studies selected were both untargeted and targeted metabolomics studies and this is indicated for each study.  In papers where both discovery and validation cohorts were employed, results from the discovery set were used for our analysis.” A small table listing some of the study characteristics and patient info has been included before the Metabolite tables. Below is the first example.

Table 1. Description of patient cohorts and study types used in GC studies

184 GC/208 HC

Unmatched

Case control study

Untargeted metabolomics

20 GC/19 HC

Unmatched

68/43 GC/HC age

8 F/12 M (GC)

Targeted metabolomics panel of 216

metabolites

84 GC/82 non-GC

Unmatched

28-79 age GC

25-82 age non-GC

45 M/39 F (GC)

Targeted metabolomics

panel of Amino acids

104 GC/50 HC

Unmatched

Untargeted

metabolomics

We would also like to state that our purpose was not to judge one paper’s techniques as being better or suboptimal.  All of these papers have been through peer review and have been published so we believe that experiments were done and analyzed properly. However, differences in techniques can cause differences in the results.  In spite of this, there were still some metabolites that were unanimously or close to unanimously agreed upon that were detected.  We have added this paragraph after Table2.  “As seen in the above table there is a large disparity between the results reported for the four different studies.  Two of the studies were untargeted and two were targeted.  This is one reason for differences as only certain metabolites are analyzed for in a targeted study. Assay conditions that differed between the studies included different extraction solvents, different columns, different HPLC gradients and different derivitization techniques. All of these could reasonably be expected to yield some differences in results. However, in spite of differences in study design and techniques, there are still metabolites which were consistently reported by all or ¾ of the studies (Table 4).”

Reviewer 2 Report

In the manuscript entitled "The distinctive serum metabolomes of gastric, esophageal and colorectal cancers" Ren et al.  elegantly present a comprehensive review of serum metabolomic profiling of GI cancers. 

Major point: 

More recently, there is a surge of identifying novel roles of gut microbiota and metabolites produced by them. Please include a section reviewing the role of microbial metabolites in these GI cancer. 

Minor point: 

  1. Lines 198-199: Please include more recent citations in support of the statement. 
  2. Line 202: From our discussion thus far, it can be seen that at the molecular level, different mutations in different tumor types occur;  please rephrase to a more meaningful sentence. 
  3. Figure 2 & 3: Consistent with Figure 1, please include the relevant microbes in these figures. This is particularly important for Figure 3 (CRC pathogenesis). 
  4. Figure 4: presentation is not very legible. Please update with a larger font size and relevant editing for better conveyance. 

Author Response

In the manuscript entitled "The distinctive serum metabolomes of gastric, esophageal and colorectal cancers" Ren et al.  elegantly present a comprehensive review of serum metabolomic profiling of GI cancers. 

Major point: 

Point 1: More recently, there is a surge of identifying novel roles of gut microbiota and metabolites produced by them. Please include a section reviewing the role of microbial metabolites in these GI cancer. 

Response 1:The last section before the Summary is a section about microbial metabolites and their relevance to cancer. We wrote the following:

“The role of gut microbiota produced metabolites in GI cancer.

Increase in certain microbial metabolites also has a role in GI cancer.  As we have just pointed out for CRC, increased amounts of lactate derived from lactic acid producing bacteria are able to cause upregulation of the MCT1 transporter to allow increased amounts of lactate into the cancer cell where it acts as a tumor promoting substance [90].  Lactate-derived pyruvate can stabilize hypoxia-inducible factor-1 (HIF-1) by inhibiting HIF polyhydroxylases which in tumor endothelial cells, stimulates angiogenesis [91]. Lactate also mediates M2-like polarization of tumor associated macrophages contributing to immunosuppression and immune escape of cancer cells [92]. 

While butyrate has been shown to have beneficial effects, in the context of APC mutations as is found in nearly all of CRC, butyrate was shown to promote proliferation of aberrant epithelial cells contributing to increased cancer polyp formation [93]. Acetaldehyde, especially in saliva is mainly of microbial source and contributes to upper GI cancer by causing molecular damage and mutagenesis [94].  The esophageal microbiome is a reflection of the oral microbiome [95] and oral pathogens have been implicated in CRC as well [96].

Secondary and tertiary bile acids (BAs) are bacterial metabolites that have been implicated in GI carcinogenesis[97] In a recent study of human BA reflux gastritis, it was found that in patients with BA reflux gastritis that there were higher amounts of conjugated primary and secondary BAs, notably, glycocholic acid (GCA), gly-cochenodeoxycholic acid (GCDCA), glycodeoxycholic acid (GDCA), taurodeoxycholic acid (TDCA), taurocholic acid (TCA) and taurochenodeoxycholic acid (TCDCA) in their gastric juice  In contrast, normal and gastritis patients without BA reflux had equal amounts of conjugated and unconjugated BAs in their gastric juice [98]”

Point 2: Minor point: 

  1. Point 2-1: Lines 198-199: Please include more recent citations in support of the statement. 

Response 2-1: We have included two additional references that are more current for the Warburg effect in cancer.

  1. Liberti MV, Locasale JW: The Warburg Effect: How Does it Benefit Cancer Cells? Trends Biochem Sci 2016, 41(3):211-218.
  2. Pascale RM, Calvisi DF, Simile MM, Feo CF, Feo F: The Warburg Effect 97 Years after Its Discovery. Cancers (Basel) 2020, 12(10).

  1. Point 2-2: Line 202: From our discussion thus far, it can be seen that at the molecular level, different mutations in different tumor typesoccur;  please rephrase to a more meaningful sentence.

Response 2-2: We have rephrased the sentence of concern in the following: “From our discussion thus far, it can be seen that at the molecular level, gene mutations are distinct and impact the phenotypes of these tumors.  Therefore, genomic re-programming in each cancer type influences its metabolome.”

  1. Point 2-3: Figure 2 & 3: Consistent with Figure 1, please include the relevant microbes in these figures. This is particularly important for Figure 3 (CRC pathogenesis). 

Response 2-3: We have added the relevant microbes to Figures 1-3.

  1. Point 2-4: Figure 4: presentation is not very legible. Please update with a larger font size and relevant editing for better conveyance. 

Response 2-4: In order to make these Impact maps more readable we have drawn them as separate Figures. 4, 5, and 6.  They have been relocated such that each Fig is included in the section that discusses the individual cancer related.

Thank you for reviewing our manuscript.  All of your suggestions have improved our manuscript.

Reviewer 3 Report

In the first part of the manuscript, Ren et al. describe the clinical subtypes and classifications of gastric and colorectal cancers in detail. The following parts, and main focus of the manuscript, reviews metabolomics papers published within the past five years for the described cancer types, restricted to LC/GC-MS based metabolite analysis in human blood samples. It is clear that the area of expertise for the authors are within the field of gastric and colorectal cancers, rather than metabolomics as the metabolomics part is described with little details and merely summarizes final results in tables with higher/lower levels of a particular metabolite.

Suggestions:

1) The authors do not compare or describe the underlying differences in pre-analytical sample collection, analytical study design and statistical data processing, which many times can explain differences and variations in results between studies. The authors need to make an effort to describe these differences for each study, instead of claiming “poor reproducibility overall between labs” as stated on line 266-268.

2) Tables 1-7 do not give any information about the analyzed samples. It is unclear if the data comes from A) LC-MS or GC-MS analysis, B) Sample matrix; serum or plasma C) Sample collection procedure, EDTA-, heparin, Na-citrate plasma etc. D) study subject characteristics (age, gender, fasting status, matching factors, diagnoses, treatment status etc.). Comparison of analytical setup is as important as evaluation of the results, as this can explain variation. The authors need to summarize this information so that the reader gets the whole picture, without having to dig through the original papers.

Author Response

Point 1: In the first part of the manuscript, Ren et al. describe the clinical subtypes and classifications of gastric and colorectal cancers in detail. The following parts, and main focus of the manuscript, reviews metabolomics papers published within the past five years for the described cancer types, restricted to LC/GC-MS based metabolite analysis in human blood samples. It is clear that the area of expertise for the authors are within the field of gastric and colorectal cancers, rather than metabolomics as the metabolomics part is described with little details and merely summarizes final results in tables with higher/lower levels of a particular metabolite.

Response 1: Thank you very much for reviewing our manuscript.  Your comments were helpful and made us look at our paper in a better way. The revised version now reads better than before.

Point 2: Suggestions:

  • Point 2-1: The authors do not compare or describe the underlying differences in pre-analytical sample collection, analytical study design and statistical data processing, which many times can explain differences and variations in results between studies. The authors need to make an effort to describe these differences for each study, instead of claiming “poor reproducibility overall between labs” as stated on line 266-268.

Response 2-1: We agree with your comments and would like to fully acknowledge that all of these papers have been through peer review and have been published so we believe that experiments were done and analyzed properly.  We have provided a small table giving a brief description of the study design before each cancer section.  Here is an example for GC.

Table 1. Description of patient cohorts and study types used in GC studies

Reference

[46]

Reference [10]

Reference [2]

Reference [48]

184 GC/208 HC

Unmatched

Case control study

Untargeted metabolomics

20 GC/19 HC

Unmatched

68/43 GC/HC age

8 F/12 M (GC)

Targeted metabolomics panel of 216

metabolites

84 GC/82 non-GC

Unmatched

28-79 age GC

25-82 age non-GC

45 M/39 F (GC)

Targeted metabolomics

panel of Amino acids

104 GC/50 HC

Unmatched

Untargeted

metabolomics

  • Point 2-2 Tables 1-7 do not give any information about the analyzed samples. It is unclear if the data comes from A) LC-MS or GC-MS analysis, B) Sample matrix; serum or plasma C) Sample collection procedure, EDTA-, heparin, Na-citrate plasma etc. D) study subject characteristics (age, gender, fasting status, matching factors, diagnoses, treatment status etc.). Comparison of analytical setup is as important as evaluation of the results, as this can explain variation. The authors need to summarize this information so that the reader gets the whole picture, without having to dig through the original papers.

Response 2-2: We would also like to state that our purpose was not to judge one paper’s techniques as being better or suboptimal.  We did not believe it appropriate to dissect each lab’s technique.  All studies were LC-MS unless otherwise stated.

We did add a bit of discussion that describes some of the differences between the studies. “As seen in the above table there is a large disparity between the results reported for the four different studies.  Two of the studies were untargeted and two were targeted.  This is one reason for differences as only certain metabolites are analyzed for in a targeted study. Assay conditions that differed between the studies included different extraction solvents, different columns, different HPLC gradients and different derivitization techniques. All of these could reasonably be expected to yield some differences in results. However, in spite of differences in study design and techniques, there are still metabolites which were consistently reported by all or ¾ of the studies (Table 4).”  This was for GC and similar passages were composed for the other two cancers.

Reviewer 4 Report

I think the paper is of interest but hardly affected by these confusing and huge tables, not readable at all. It makes no sense to draw these tables, perhaps you can list only the increased/decreased aminoacids or the most significant and not the complete list, perhaps using rows and not columns. 

Conclusions are poor, a discussion is needed and should touch at least these points:

  • correlation between different subtypes of cancer and metabolic profile
  • correlation between metabolic profile and prognosis
  • correlation between metabolic profile and response to therapy
  • future perspectives

Author Response

Point 1: I think the paper is of interest but hardly affected by these confusing and huge tables, not readable at all. It makes no sense to draw these tables, perhaps you can list only the increased/decreased aminoacids or the most significant and not the complete list, perhaps using rows and not columns. 

Response 1: Thank you for your comments. Except for the first long table which illustrates how we figured out which metabolites were the most consistently reported, the other two very long tables have been placed into a SI file. Only the most significant metabolites are now tabulated in the manuscript.

Conclusions are poor, a discussion is needed and should touch at least these points:

  • Point 2: correlation between different subtypes of cancer and metabolic profile

Response 2: Our analysis was only comparing cancer to non-cancer patients. These studies did not touch on stages of cancer, even though different stages of cancer were present. The metabolite profile was just for CRC vs non-CRC, etc.  We did add some extensive discussion about the relevance to cancer for the metabolites that came from our selection technique.

For example, for GC we wrote the following. The oxidative phosphorylation of glucose is impaired in the mitochondria of cancer cells as a result of the Warburg effect and thus, the number of the acetyl-CoA molecules derived from glucose is significantly reduced. Instead, cancer cells rely on upregulation of amino acid biosynthesis and metabolism to replenish TCA cycle intermediates to generate ATP [2, 51]. Trp catabolism is important for both the production of Acetyl-CoA for use in the TCA cycle and both His and Trp are used for the anabolism of one-carbon units for the synthesis of nucleotides for DNA and RNA biosynthesis [52]. Kynurenine, a major metabolite of Trp via the enzyme indoleamine-2,3-dioxygenase (IDO1), induces immunosuppression by binding to and activating the transcription factor aryl hydrocarbon receptor (AhR) [53, 54]. This inhibits the ability of immune-tolerant dendritic cells (DCs) and regulatory T cells to target and eliminate cancer cells [55, 56].

Glu is also used by cancer cells in the TCA cycle and it can be biosynthesized by the transamination of His and Arg and/or by the deamination of glutamine [51, 57]. Thr catabolism via Thr dehydrogenase produces Gly and acetyl-CoA which can also feed the TCA cycle [58]. Met can also enter the Met-folate cycle to produce 1-carbon units for nucleobases [59]. All of the above mentioned amino acids were found to be reduced in the serum of GC patients possibly due to the upregulation of these metabolic pathways in the tumor. The Arg-Ornithine-polyamine pathway may be used to synthesize polyamines that promote the proliferation of cancer cells. Ornithine can be converted to citrulline in the ureacycle to replenish Arg supplies as Arg is decreased in most tumors [57, 60, 61]. Tyr is important for integration into proteins that activate important oncogenic signaling pathways such as Kras [62].

  • Point 3: correlation between metabolic profile and prognosis

Response 3: The prognosis information was not always available but we included it when possible.  For EC we wrote: “One metabolic alteration in cancer is the accumulation of free fatty acids (FFAs) which enhances proliferation [72]. Fas-associated actor-1 (FAF1) contains a binding motif for unsaturated FAS but not for saturated FAs such as palmitic acid. FAF1 is a protein that facilitates the degradation of β-catenin [73]. Oleic acid and linoleic acid were both found to bind to FAF1 in various cancer cell lines and stabilize β-catenin, a transcriptional co-activator that stimulates expression of genes that drive cell proliferation [74]. Palmitoleic acid is an unsaturated FA which can be biotransformed to palmitic acid which can then be used to produce more oleic acid*. Palmitoleic acid can also be produced from palmitic acid by Δ9 desaturase [75].  Increased expression of β-catenin has been observed in ESCC and was correlated with a poor patient prognosis [76].  Elevated linoleic acid metabolism has been previously reported in EC patients [77].”

  • Point 4: correlation between metabolic profile and response to therapy

Response 4: We did include this when it was available. For CRC we wrote:

“The metabolites which have not been discussed and are distinct for CRC are Phe, Cys, and lactic acid.  Phe is important for the biosynthesis of Tyr whose importance to cancer cells has already been discussed extensively in the section related to GC [57]. Significant amount of ROS are generated in cancer cells due to their increased proliferation and oxidative stress and can cause cell death. Therefore, cancer cells must have increased antioxidant defenses to neutralize their increased ROS production. Glutathione is essential in maintaining redox balance in all subcellular compartments [59].  Production of glutathione requires Glu, Gly, and Cys with Cys being the most critical component because of its thiol group. Inhibiting cysteine uptake has been shown to reduce cancer cell viability which was caused by uncontrolled oxidative stresses [82]. Cysteine can be imported into cells either directly or in its oxidized form, cystine, through the cystine/glutamate antiporter system xc− (xCT) [59].  Studies have looked at the efficacy of xCT inhibitors for cancer treatment [83, 84].  Lactic acid is the end-product of glycolysis and would be expected to raise higher for cancer patients than HCs [57, 75].  CRC Patients with higher serum Lactic acid levels were found to have poor prognosis, especially for metastatic CRCb [85]  Exogenous lactate derived from metabolism of lactic acid producing bacteria in the gut can serve as a fuel source for oxidative cancer cells and cause upregulation of monocarboxylate transporter 1 (MCT1).  Therefore, lactate is considered a tumor promoting metabolite as it can influence angiogenesis, amino acid metabolism, histone deacetlyases and immune escape contributing to cell migration [86].”

  • Point 5: future perspectives

Response 5: We wrote this for our summary and future perspective.

“In this review we sought those biomarkers that were present in most of the studies. If a differential metabolite was reported ≥70% of the time by different labs in different parts of the world in different years, then we felt this could not be a coincidence.  All of the biomarkers selected in this review were highly relevant to cancer metabolism and were highlighted in a discussion following each cancer section of the review. Metabolomics shows promise for cancer screening but a more cooperative effort between labs must be established to standardize assays and study designs for this technique to be used in the clinic such that on any given day in any part of the world a patient will receive an accurate assessment of his metabolite profile.  The fact that very different types of metabolites were detected by different labs with relatively low reproducibility is a good indication that assay conditions were very different for each of the labs. While differences in technique and study design can explain differences in results, a true biomarker should be directly keyed to the disease.  In other words, if the tumor type is present then the differential biomarker should be there.  Only if protocols become uniform can this type of biomarker be truly discovered and validated. Independent labs that use their time and resources to repeat and validate a published assay should perhaps be given publication opportunities as this is important for the metabolomics community. Thus, the publication industry has a big role to play as publication should be used to disseminate scientific findings that are not just unique but also for the benefit of the public, in this case, cancer patients.  Allowing other scientists to read about good and reproducible results will encourage the use of those assays and produce more consistent findings that will eventually lead to a trustworthy clinical assay.  Funding agencies should also provide help to those labs which are willing to perform assay validation work.  Once consistent biomarkers are found, metabolomics labs will develop more targeted assays to provide information about cancer vulnerabilities and enable a more selective targeting of a metabolic pathway while preserving as much normal physiological function as possible. During this process we shall also learn more about why established drugs work better for some cancers. If the ideas put forth here that metabolomic assays must eventually become standardized and uniform are seriously considered as worthwhile, we believe that having metabolomics assays implemented in the clinical lab is still winnable.”

Round 2

Reviewer 1 Report

Dear Authors,

thank you for addressing my comments and including the heatmap, although I do not understand why you replaced only one table with a heatmap. It is so much easier to understand.

Now that I have seen and understood that you had 2 targeted and 2 untargeted platforms in the study I am surprised you find any overlap at all. Especially since one study only investigated amino acids.

reviewer

Reviewer 2 Report

The manuscript has significantly improved after all the updates. In sincerely thank and congratulate the authors for the excellent work. 

Reviewer 4 Report

Paper significantly improved and suitable for publication